# Surgical Margins Status and Prognosis after Resection of Oral Cavity Squamous Cell Carcinoma: Results from a Taiwanese Nationwide Registry-Based Study

**DOI:** 10.3390/cancers14010015

**Published:** 2021-12-21

**Authors:** Chung-Jan Kang, Yu-Wen Wen, Shu-Ru Lee, Li-Yu Lee, Chuen Hsueh, Chien-Yu Lin, Kang-Hsing Fan, Hung-Ming Wang, Chia-Hsun Hsieh, Shu-Hang Ng, Chih-Hua Yeh, Chih-Hung Lin, Chung-Kan Tsao, Tuan-Jen Fang, Shiang-Fu Huang, Li-Ang Lee, Ku-Hao Fang, Tzu-Chen Yen, Nai-Ming Cheng, Tsung-You Tsai, Shiao-Fwu Tai, Chi-Ying Tsai

**Affiliations:** 1Department of Otorhinolaryngology, Head and Neck Surgery, Chang Gung Memorial Hospital, Medical College of Chang Gung University, Taoyuan 333, Taiwan; keny@cgmh.org.tw (C.-J.K.); fang3109@adm.cgmh.org.tw (T.-J.F.); bigmac@cgmh.org.tw (S.-F.H.); 5738@cgmh.org.tw (L.-A.L.); b8401085@cgmh.org.tw (K.-H.F.); mp0594@cgmh.org.tw (T.-Y.T.); xiaofu@cgmh.org.tw (S.-F.T.); 2Clinical Informatics and Medical Statistics Research Center, Medical College of Chang Gung University, Taoyuan 333, Taiwan; ywwen@mail.cgu.edu.tw; 3Division of Thoracic Surgery, Chang Gung Memorial Hospital, Taoyuan 333, Taiwan; 4Research Service Center for Health Information, Chang Gung University, Taoyuan 333, Taiwan; s19880711@mail.cgu.edu.tw; 5Department of Pathology, Chang Gung Memorial Hospital, Medical College of Chang Gung University, Taoyuan 333, Taiwan; r22068@cgmh.org.tw (L.-Y.L.); ch9211@adm.cgmh.org.tw (C.H.); 6Department of Radiation Oncology, Chang Gung Memorial Hospital, Medical College of Chang Gung University, Taoyuan 333, Taiwan; qqvirus@cgmh.org.tw (C.-Y.L.); kanghsing@cgmh.org.tw (K.-H.F.); 7Department of Medical Oncology, Chang Gung Memorial Hospital, Medical College of Chang Gung University, Taoyuan 333, Taiwan; whm526@cgmh.org.tw (H.-M.W.); wisdom500@cgmh.org.tw (C.-H.H.); 8Department of Diagnostic Radiology, Chang Gung Memorial Hospital, Medical College of Chang Gung University, Taoyuan 333, Taiwan; shng@cgmh.org.tw (S.-H.N.); foliatus@cgmh.org.tw (C.-H.Y.); 9Department of Plastic and Reconstructive Surgery, Chang Gung Memorial Hospital, Medical College of Chang Gung University, Taoyuan 333, Taiwan; chihhung@cgmh.org.tw (C.-H.L.); nightman@adm.cgmh.org.tw (C.-K.T.); 10Department of Nuclear Medicine and Molecular Imaging Center, Chang Gung Memorial Hospital, Medical College of Chang Gung University, Taoyuan 333, Taiwan; yen1110@cgmh.org.tw (T.-C.Y.); Tc99m@cgmh.org.tw (N.-M.C.); 11Department of Oral and Maxillofacial Surgery, Chang Gung Memorial Hospital, Medical College of Chang Gung University, Taoyuan 333, Taiwan

**Keywords:** oral cavity squamous cell carcinoma, surgical margin, outcome, local control, nationwide registry study

## Abstract

**Simple Summary:**

While the prognostic role of surgical margins in oral cavity squamous cell carcinoma is well-established, the optimal cutoff values for margin status remain controversial. This study addressed this issue in a large sample of 13,768 patients included in a nationwide registry in Taiwan. The identification of the most suitable cutoff value for surgical margins was conducted using 5-year local control as the outcome of interest. On analyzing the margin status—categorized as 0, 0.1–4 and > 4 mm—the 5-year outcomes were as follows: local control, 87, 89 and 92%; disease-specific survival, 57, 76 and 81%; overall survival, 47, 67 and 74%, respectively. Collectively, these data indicate that a margin status >4 mm can be considered as adequate.

**Abstract:**

(1) Background: The optimal cutoff value that maximizes the prognostic value of surgical margins in patients with resected oral cavity squamous cell carcinoma has not yet been identified. (2) Methods: Data for this study were retrieved from the Taiwan Cancer Registry Database. A total of 13,768 Taiwanese patients with oral cavity squamous cell carcinoma were identified and stratified according to different margin statuses (0, 0.1–4 and > 4 mm). The five-year local control, disease-specific survival and overall survival rates were the main outcome measures. (3) Results: The 5-year local control, disease-specific survival and overall survival rates of patients with close margins (0 and 0.1–4 mm) were significantly lower than those observed in patients with clear margins (> 4 mm; all *p* values < 0.001). In multivariate analysis, margin status, depth of invasion and extra-nodal extension were identified as independent adverse prognostic factors for 5-year local control. (4) Conclusions: A thorough assessment of surgical margins can provide a reliable prognostic prediction in patients with OCSCC. This has potential implications for treatment approaches tailored to the individual level. The achievement of clear margins (>4 mm) should be considered a key surgical goal to improve outcomes in this patient group.

## 1. Introduction

With only 5% of oral malignancies being other types, the most common form of oral cancer is oral cavity squamous cell carcinoma (OCSCC). Estimates derived from epidemiological data have shown that the incidence rates of OCSCC are rapidly increasing worldwide. While surgical excision remains the mainstay of therapy, several studies have reported that post-operative adjuvant concurrent chemoradiotherapy (CCRT) or radiotherapy (RT) alone may improve prognosis in patients with OCSCC [1,2]. Importantly, a suboptimal tumor resection with an inadequate tumor resection margin portends adverse outcomes, including reduced survival and an increased risk of recurrences.

The NCCN guidelines (version: February 2021) for head and neck malignancies maintain that a surgical margin should be considered clear when the distance from the tumor invasion front is ≥ 5 mm [3]. Historically, the association between positive margins and poor outcomes in OCSCC was described for the first time in 1978 [4]. Thereafter, Chen et al. [5] proposed a cutoff of 5 mm for clear surgical margins—which is in line with the current NCCN guidelines.

While the prognostic role of surgical margins in OCSCC is well-established, the optimal cutoff values have not been thoroughly assessed, especially with respect to different endpoints (e.g., local control (LC), disease-specific survival (DSS) and overall survival (OS). In the present study, we aimed to address this knowledge gap using Taiwan’s nationwide data from all patients diagnosed with OCSCC in the country. The analysis of different cutoff points for surgical margins was conducted with respect to the 5-year LC rate.

## 2. Materials and Methods

### 2.1. Patient Selection

Ethics approval for the study was received from the Institutional Review Board (IRB: 201801398B0A3) of the Chung Gung Medical Foundation. A waiver for informed consent was obtained because data were collected from a nationwide registry. This is a retrospective analysis of prospectively collected data obtained from the Taiwan Cancer Registry Database (TCRD). The registry is administered by the Taiwanese Ministry of Health and Welfare through a state agency (Health and Welfare Data Science Center). The TCDR has prospectively recorded information on > 98% of all patients with OCSCC in Taiwan, along with their outcomes. Patients included in the TCDR who met the ICDO-3 diagnostic criteria for tongue cancer, lip cancer, floor of mouth cancer, alveolar ridge cancer, buccal cancer, retromolar trigone cancer hard palate cancer and other types of oral cancer were deemed eligible. The enrollment period started in January 2011 and concluded in December 2017. Follow-up was continued until December 2019. A study flowchart is provided in Figure 1. The following exclusion criteria were applied: presence of an in situ carcinoma (*n* = 165), known history of other malignancies (*n* = 8741), initial non-surgical treatment (*n* = 4425), unavailable data on pathological stage (*n* = 531), unknown surgical margins or extranodal extension (*n* = 3867), unknown depth of invasion, unknown lymph node status or tumor differentiation (*n* = 3163) and margins > 30 mm (*n* = 8). After these exclusions, the study sample consisted of 13,768 patients (Figure 1).

### 2.2. Statistical Analysis

All calculations were performed with SAS, version 9.4 (SAS Institute Inc., Cary, NC, USA) and the R statistical package (version 4.0.2). During the study period, data on five-year local control (LC), disease-specific survival (DSS) and overall survival (OS) were acquired from the TCRD. Mortality data were confirmed using the Taiwanese National Register of Deaths. Kaplan-Meier curves for different endpoints were plotted and intergroup differences were examined using the log-rank test. We estimated hazard ratios (HRs) and 95% confidence intervals (CIs) for the associations between risk factors and the study endpoints by means of univariate and multivariate Cox proportional hazards regression models. All variables included in univariate analysis were entered into the final model and a stepwise selection procedure was applied. Statistical significance was determined by a *p* value < 0.05 (two-tailed).

## 3. Results

### 3.1. Patient Characteristics in Relation to Their Margin Status

In total, we evaluated 13,768 patients (12,548 men and 1220 women) with OCSCC. In the subgroup of patients who had been treated with surgery alone, the optimal cutoff value when 5-year LC was used as the outcome of interest was 5 mm. However, when the analysis was extended to include patients who had undergone postoperative adjuvant therapy, an optimal cutoff point of 3 mm was identified. In the entire study cohort, the optimal cutoff value according to the results of Cox proportional hazards analysis was 4 mm. The study dataset was therefore analyzed using two different cutoffs (i.e., 4 mm and 5 mm). In multivariate analysis, margin status was identified as an independent risk factor for both 5-year local control and disease-specific survival regardless of the cutoff value used for calculations. However, margin status was retained in the model as an independent risk factor for 5-year overall survival only when a cutoff of 4 mm was used. On analyzing the prognostic significance of the margin status using different cutoff values (from 0–30 mm), we subsequently applied Cox proportional hazard models with a spline function using 5-year LC as the outcome of interest. A margin of 4 mm produced a HR of 10,245; therefore, this value was selected for subsequent analyses. Table 1 shows the patient characteristics in relation to margin status (0, 0.1–4 or > 4 mm). The following variables showed significant intergroup differences (all *p* < 0.001 for 0 and 0.1–4 versus > 4 mm): pT1 (7.5 versus 20.8 and 21.5%, respectively), pT2 (19.3 versus 32.9 and 33.4%, respectively), pN0 (46.7 versus 65.5 and 70.9%, respectively), p-stage I (5.9 versus 18.0 and 19.0%, respectively), p-stage II (10.4 versus 22.8 and 25.2%, respectively), depth of invasion < 10 mm (33.6 versus 54.3 and 54.6%, respectively), absence of extra-nodal extension (ENE) (69.5 versus 84.1 and 86.5%, respectively) and treatment with surgery alone (20.0 versus 46.9 and 54.4%, respectively). We found no significant intergroup differences in terms of sex, age and presence of second primary cancers.

### 3.2. Five-Year Survival Rates in Relation to the Patients’ Margin Statuses

The 5-year outcomes in relation to the patients’ margin statuses (0, 0.1–4 and > 4 mm) were as follows: LC, 87, 89 and 92%, respectively (log-rank test, *p* < 0.001); DSS, 57, 76 and 81%, respectively (log-rank test, *p* < 0.001); and OS, 47, 67 and 74%, respectively (log-rank test, *p* < 0.001; Figure 2A–C). Therefore, the outcomes of patients with margins >4 mm were more favorable than those of cases with margins ≤4 mm.

### 3.3. Cox Regression Analysis

The following categories served as references (HR = 1): margin status >4 mm, age <65 years, female sex, p-stage I−II, depth of invasion <10 mm, pT1−2, pN0, absence of ENE, treatment with surgery alone, absence of second primary cancers, lip tumor site and presence of a well differentiated tumor. On univariate analyses, the following variables were significantly associated with five-year LC: margin status (0 and 0.1–4 versus >4 mm), pT classification, pN classification, p-stage, depth of invasion, ENE, treatment modality, tumor site and tumor differentiation. The same variables and age (≥65 versus <65 years) were identified as being significantly associated with 5-year DSS. Univariate analyses identified margin status (0 and 0.1–4 versus >4 mm) as significantly associated with 5-year DSS. The same variables, and the presence (versus absence) of second primary cancers, were significantly associated with 5-year OS. Univariate analyses identified margin status (0 and 0.1–4 versus > 4 mm) as significantly associated with 5-year OS (Table 2). The results of multivariate analyses identified the following risk factors as independently associated with 5-year OS: margin status (0 and 0.1–4 versus > 4 mm), sex, age, pT classification, pN classification, depth of invasion, ENE, second primary cancer, tumor site and tumor differentiation. Notably, margin status (0 and 0.1–4 versus > 4 mm), sex, age, pT classification, pN classification, p-stage, depth of invasion, depth of invasion, ENE, tumor site and tumor differentiation were identified as independently associated with five-year DSS. Moreover, margin status (0 and 0.1–4 versus > 4 mm), depth of invasion, tumor site and tumor differentiation showed independent associations with 5-year LC (Table 2).

## 4. Discussion

By taking advantage of the TCRD dataset, we conducted a nationwide analysis of the prognostic role played by margin status in Taiwanese patients with OCSCC who had undergone surgical resection. Notably, the TCRD offers highly comprehensive coverage (>98%) of OCSCC cases occurring in Taiwan. The Surveillance, Epidemiology, and End Results study comprised only 9.4% of patients with OSCSS diagnosed in the United States between 1975 and 2013 [6].

The prognostic significance of surgical margins in OCSCC has been the subject of intense investigation, but a consensus on their optimal cutoff value is still lacking. The seminal retrospective study published in 1987 by Chen et al. [5] examined this issue in 270 patients who were diagnosed with head and neck carcinoma. Using a cutoff value of 5 mm, they found that the presence of an inadequate margin had an adverse prognostic significance on 5-year DFS; however, the relative weight of surgical margins in determining clinical outcomes with respect to other variables was not specifically examined. More recently, two independent studies demonstrated that patients with OCSCC who had clear margins (>5 mm) had higher 5-year OS rates than those with close margins (≤5 mm) [4,7]. On analyzing the optimal cutoff value for margin status, we found that patients with margins >4 mm had a 5-year OS rate of 74%—which is higher than that reported in the published literature. As far as LC and DSS are concerned, McMahon et al. [8] identified the margin status (cutoff: 10 mm) as a univariate predictor for both endpoints. In this study, patients with OCSCC who had clear margins (>4 mm) had higher 5-year LC and DSS rates than those with close margins (≤4 mm). Collectively, these data indicate that a cutoff of 4 mm for margin status has clinical value in the prediction of different endpoints of interest.

Surgical margins have been previously identified as a variable predicting LC in Taiwanese patients [9]. In the current study, we extended our previous observation by showing that margins predict not only 5-year LC but also DSS and OS. A similar trend has been reported by Patel et al. [10] with respect to DSS, although in their study the difference failed to reach statistical significance. On categorizing margin status into three groups (positive: 0; close ≤2 and clear >2 mm), Binahmed et al. [11] demonstrated that cases with clear margins had the lowest recurrence rate and the highest 5-year OS; these findings are in line with our current observations (Table 3). However, the independent prognostic significance of the margin status for 5-years OS was not retained in the multivariate model—possibly as a result of an inadequate cutoff value (2 mm). Using a cutoff of 5 mm, Jang et al. [12] found a higher risk of local recurrences in presence of close or positive margins compared with clear margins. Additionally, clear margins were predictive of local recurrences only in advanced disease stages (III and IV). Nason et al. [13] analyzed the prognostic significance of margin status using a four-category classification system, as follows: ≥5, 3−4, ≤2 mm and positive margins. They found that the larger the surgical margin, the higher the 5-year OS rate. Intriguingly, the recurrence rate of patients within the 3−4 mm category was slightly lower than that observed in the ≥5 mm category (24.4 versus 25%, respectively), a finding consistent with our current observations. While there were no differences with respect of 5-year DFS between the ≤2 mm (49%) and positive margins (48%) categories, the 5-year OS of the latter group was markedly lower (39%) than that observed in the former (63%). A potential explanation may lie in the fact that patients with margins ≤2 mm were actively treated with adjuvant RT or CCRT in the post-surgical phase. On analyzing the prognostic significance of margin status using different categories and a cutoff point of 5 mm, Tasche et al. [14] and Zanoni et al. [15] found the lowest recurrence risk in presence of clear margins (>5 mm). Additionally, the latter research group assessed the impact of margins on 2-year local recurrence-free survival and failed to identify differences between the 2.3–5 mm category and the clear margins (>5 mm) category. [15] They therefore, concluded that a cutoff of 2.2 mm may be clinically useful for prognostic stratification.

In 2003, Sutton et al. [16] conducted a study in 200 patients with OCSCC in whom margin status was classified using the following three-category system: clear (≥5 mm), close (<5 mm) and positive (0 mm). The 5-year local recurrence and OS rates in the three margin categories were 12%/33%/55% and 54%/26%/0%, respectively. A similar association with local recurrences and OS was also reported by Loree and Strong [4]. Previous studies from our group demonstrated that the survival outcomes of patients with T4a and T4b malignancies of the oral cavity—which were deemed unresectable according to the AJCC 2002 staging manual—are not uniform [17,18]. These results prompted a revision of the resectability guidelines, with the AJCC 2010 staging manual considering T4a tumors amenable to excision [19].

On analyzing patients with OCSCC who had been treated with surgery alone, we identified an optimal cutoff value of 5 mm. However, when the analysis was extended to include patients who had undergone postoperative adjuvant therapy, the optimal cutoff point was 3 mm. These results indicate that patients with OCSCC and close margins should undergo more aggressive post-operative treatment modalities. The current study expands our previous observations by showing that the prognosis of patients with margins >4 mm is favorable both in terms of LC and OS, ultimately providing a solid foundation for a further prognostic refinement and tailored treatment.

Using nationwide data from Taiwan, Lin et al. [20] recently analyzed the association between the status of surgical margins and survival outcomes in patients with OCSCC. They found that cases with margins ≥4 or 5 mm showed more favorable 5-year cancer-specific survival (CSS) and OS rates. Our current investigation has several methodological differences compared with the study by Lin and coworkers [20]. First, only patients with first primary OCSCC were included in our research. Second, we did not include patients with disease recurrences, who are known for having a distinct pattern of risk factors and a less favorable clinical course. Third, the optimal cutoffs for margin status were determined by applying multivariate Cox regression analyses using cancer-specific survival and OS the endpoints of interest. Here, we chiefly focused on LC as a key driver of clinical outcomes; additionally, we tested different cutoff values (from 0–30 mm) using a spline approach with the goal of identifying the point that maximized the HR for the prediction of LC. Fourth, differently from Lin and colleagues, we excluded patients with unknown data on second primary cancers and pNx, resulting in a lower sample size. Despite these differences, the two studies consistently demonstrated that the achievement of clear margins (current study: >4 mm; Lin et al.: ≥5 mm) is a key surgical goal to improve outcomes. Subject to independent confirmation, we believe that surgeons in charge of OCSCC excision should aim to achieve a margin >4 mm.

There are limitations to this study. First, our investigation was conducted in an area where betel quid chewing is endemic and the external validity of our findings in Western countries deserves further scrutiny. Second, for patients with OCSCC, there are many factors that could have had an effect on prognosis that could mitigate the effect of surgical margin status on clinical outcomes [21,22]. Third, the results of registry-based studies can be affected by unmeasured confounding or residual confounding.

## 5. Conclusions

In conclusion, we have shown that thorough assessments of surgical margins can provide reliable prognostic predictions for patients with OCSCC, with potential implications for treatment approaches tailored to the individual level. The achievement of clear margins (>4 mm) should be considered as a key surgical goal to improve outcomes in this patient group.

## Figures and Tables

**Figure 1 cancers-14-00015-f001:**
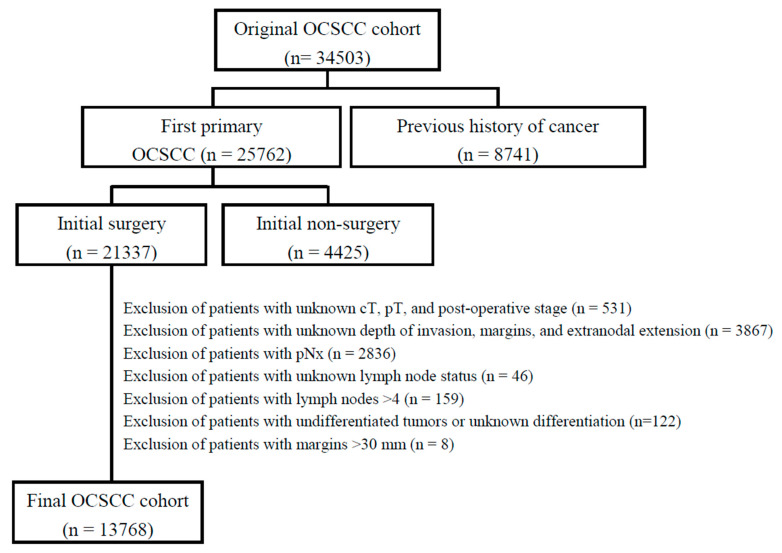
Flow of patients through the study. OCSCC, oral squamous cell carcinoma.

**Figure 2 cancers-14-00015-f002:**
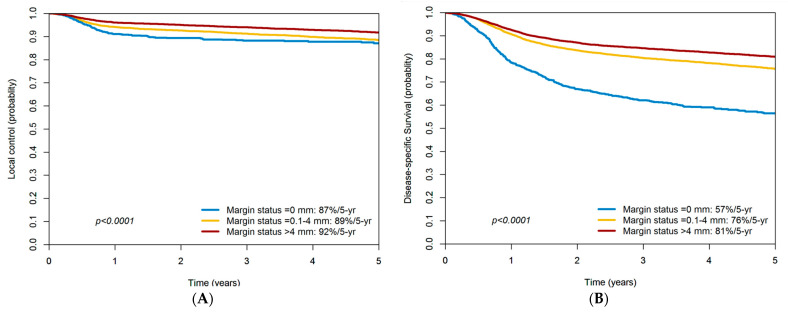
(**A**) Five-year local control rates of patients with oral cavity squamous cell carcinoma stratified according to different margin statuses (0, 0.1–4 and > 4 mm); (**B**) five-year disease-specific survival rates of patients with oral cavity squamous cell carcinoma stratified according to different margin statuses (0, 0.1–4 and > 4 mm); (**C**) five-year overall survival rates of patients with oral cavity squamous cell carcinoma stratified according to different margin statuses (0, 0.1–4 and > 4 mm).

**Table 1 cancers-14-00015-t001:** General characteristics of patients with oral cavity squamous cell carcinoma stratified according to different margin statuses (0, 0.1–4 and > 4 mm).

Characteristic	Margin Status 0 mm (*n* = 825)	Margin Status 0.1–4 mm (*n* = 6105)	Margin Status >4 mm (*n* = 6838)	*p*
Sex (No., %)				
Male (12,548, 91.1)	760 (92.1)	5548 (90.9)	6240 (91.3)	0.4447
Female (1220, 8.9)	65 (7.9)	557 (9.1)	598 (8.7)	
Age, years (No., %)				
<65 (11,733, 85.2)	695 (84.2)	5194 (85.1)	5844 (85.5)	0.5927
≥65 (2035, 14.8)	130 (15.8)	911 (14.9)	994 (14.5)	
Pathologic T status (No., %)				
T1 (2799, 20.3)	62 (7.5)	1270 (20.8)	1467 (21.5)	<0.0001
T2 (4454, 32.4)	159 (19.3)	2008 (32.9)	2287 (33.4)	
T3 (2094, 15.2)	137 (16.6)	956 (15.7)	1001 (14.6)	
T4 (4421, 32.1)	467 (56.6)	1871 (30.6)	2083 (30.5)	
Pathologic N status (No., %)				
pN0 (9227, 67.0)	385 (46.7)	3996 (65.5)	4846 (70.9)	<0.0001
pN1 (1303, 9.5)	98 (11.9)	618 (10.1)	587 (8.6)	
pN2 (1487, 10.8)	124 (15.0)	697 (11.4)	666 (9.7)	
pN3 (1751, 12.7)	218 (26.4)	794 (13.0)	739 (10.8)	
Pathologic stage (No., %)				
I (2449, 17.8)	49 (5.9)	1098 (18.0)	1302 (19.0)	<0.0001
II (3198, 23.2)	86 (10.4)	1389 (22.8)	1723 (25.2)	
III (2108, 15.3)	109 (13.2)	996 (16.3)	1003 (14.7)	
IV (6013, 43.7)	581 (70.5)	2622 (42.9)	2810 (41.1)	
Depth of invasion, (No., %), mm				
<10 (7322, 53.2)	277 (33.6)	3312 (54.3)	3733 (54.6)	<0.0001
≥10 (6446, 46.8)	548 (66.4)	2793 (45.7)	3105 (45.4)	
Extra-nodal extension (No., %)				
No (11,620, 84.4)	573 (69.5)	5133 (84.1)	5914 (86.5)	<0.0001
Yes (2148, 15.6)	252 (30.5)	972 (15.9)	924 (13.5)	
Treatment modality (No., %)				
S alone (6749, 49.0)	165 (20.0)	2861 (46.9)	3723 (54.4)	<0.0001
S plus CT+S plus RT+S plus CT and RT (7019, 51.0)	660 (80.0)	3244 (53.1)	3115 (45.6)	
Second primary tumors (No., %)				
No (12,176, 88.4)	734 (89.0)	5400 (88.5)	6042 (88.4)	0.8734
Yes (1592, 11.6)	91 (11.0)	705 (11.5)	796 (11.6)	
Site (No., %)				
Lip (596, 4.3)	20 (2.4)	223 (3.7)	353 (5.2)	<0.0001
Tongue (5252, 38.1)	198 (24.0)	2196 (36.0)	2858 (41.8)	
Gum (1803, 13.1)	174 (21.1)	792 (13.0)	837 (12.2)	
Mouth floor (511, 3.7)	45 (5.5)	260 (4.3)	206 (3.0)	
Palate (181, 1.3)	34 (4.1)	73 (1.2)	74 (1.1)	
Buccal (4463, 32.4)	267 (32.4)	2167 (35.5)	2029 (29.7)	
Retromolar (606, 4.4)	56 (6.8)	252 (4.1)	298 (4.4)	
Other sites (356, 2.7)	31 (3.7)	142 (2.2)	183 (2.6)	
Tumor differentiation (No., %)				
Well differentiated (3786, 27.5)	198 (24.0)	1648 (27.0)	1940 (28.4)	<0.0001
Moderately differentiated (8674, 63.0)	509 (61.7)	3877 (63.5)	4288 (62.7)	
Poorly differentiated (1308, 9.5)	118 (14.3)	580 (9.5)	610 (8.9)	

Other sites: tumors arising from the retromolar trigone or overlapping regions. Abbreviations: S, surgery; CT, chemotherapy; RT, radiotherapy.

**Table 2 cancers-14-00015-t002:** Univariate and multivariate analyses of different outcome measures.

Risk Factor	Five-Year Local Control				Five-Year Disease-Specific Survival				Five-Year Overall Survival			
	Univariate Analysis		Multivariate Analysis-Stepwise		Univariate Analysis		Multivariate Analysis-Stepwise		Univariate Analysis		Multivariate Analysis-Stepwise	
	HR (95% CI)	*p*	HR (95% CI)	*p*	HR (95% CI)	*p*	HR (95% CI)	*p*	HR (95% CI)	*p*	HR (95% CI)	*p*
Margin status												
>4 mm	1		1		1		1		1		1	
0.1–4 mm	1.46 (1.29–1.64)	<0.0001	1.42 (1.26–1.61)	<0.0001	1.32 (1.22–1.42)	<0.0001	1.24 (1.15–1.33)	<0.0001	1.28 (1.20–1.36)	<0.0001	1.23 (1.15–1.31)	<0.0001
0 mm	1.79 (1.44–2.23)	<0.0001	1.56 (1.25–1.95)	<0.0001	2.74 (2.43–3.08)	<0.0001	1.70 (1.51–1.92)	<0.0001	2.52 (2.27–2.79)	<0.0001	1.69 (1.52–1.88)	<0.0001
Sex												
Female	1		−	−	1		1		1		1	
Male	1.19 (0.95–1.47)	0.1263	−	−	1.05 (0.92–1.19)	0.4754	1.15 (1.01–1.33)	0.0349	1.10 (0.99–1.23)	0.0858	1.21 (1.08–1.36)	0.0008
Age, years												
<65	1		−	−	1		1		1		1	
≥65	0.91 (0.77–1.08)	0.2677	−	−	1.25 (1.14–1.38)	<0.0001	1.30 (1.18–1.43)	<0.0001	1.50 (1.39–1.62)	<0.0001	1.62 (1.50–1.75)	<0.0001
Pathologic T status												
T1-2	1		−	−	1		1		1		1	
T3-4	1.25 (1.11–1.40)	0.0001	−	−	2.77 (2.57–2.99)	<0.0001	1.38 (1.22–1.57)	<0.0001	2.47 (2.32–2.63)	<0.0001	1.49 (1.36–1.64)	<0.0001
Pathologic N status												
pN0	1		−	−	1		1		1		1	
pN1-3	1.32 (1.17–1.48)	<0.0001	−	−	4.14 (3.86–4.45)	<0.0001	2.14 (1.92–2.39)	<0.0001	3.44 (3.24–3.65)	<0.0001	2.05 (1.90–2.22)	<0.0001
Pathologic stage												
I-II	1		−	−	1		1		1		−	−
III-IV	1.34 (1.19–1.51)	<0.0001	−	−	4.31 (3.92–4.74)	<0.0001	1.32 (1.13–1.54)	0.0004	3.37 (3.13–3.63)	<0.0001	−	−
Depth of invasion												
<10 mm	1		−	−	1		1		1		1	
≥10 mm	1.29 (1.15–1.45)	<0.0001	1.22 (1.09–1.38)	0.0010	2.60 (2.42–2.81)	<0.0001	1.31 (1.18–1.46)	<0.0001	2.33 (2.19–2.48)	<0.0001	1.31 (1.19–1.43)	<0.0001
Extra-nodal extension												
No	1		−	−	1		1		1		1	
Yes	1.39 (1.20–1.60)	<0.0001	−	−	4.35 (4.04–4.68)	<0.0001	1.82 (1.66–1.99)	<0.0001	3.89 (3.65–4.15)	<0.0001	1.78 (1.64–1.93)	<0.0001
Treatment modality												
S alone	1		−	−	1		−	−	1		−	−
S plus CT + S plus RT + S plus both CT and RT	1.25 (1.11–1.40)	0.0002	−	−	2.74 (2.53–2.96)	<0.0001	−	−	2.30 (2.16–2.45)	<0.0001	−	−
Second primary												
No	1		−	−	1		−	−	1		1	
Yes	0.93 (0.78–1.12)	0.4456	−	−	1.02 (0.92–1.13)	0.7575	−	−	1.76 (1.63–1.90)	<0.0001	1.71 (1.58–1.84)	<0.0001
Site												
Lip	1		1		1		1		1		1	
Tongue	1.24 (0.88–1.77)	0.2238	1.15 (0.81–1.63)	0.4516	1.32 (1.09–1.62)	0.0056	1.01 (0.82–1.23)	0.9627	1.14 (0.98–1.34)	0.0995	0.92 (0.78–1.08)	0.3140
Gum	1.45 (0.99–2.10)	0.0538	1.30 (0.90–1.89)	0.1677	1.63 (1.32–2.02)	<0.0001	1.14 (0.92–1.41)	0.2312	1.35 (1.14–1.60)	0.0005	0.99 (0.83–1.17)	0.8679
Mouth floor	0.83 (0.49–1.40)	0.4781	0.72 (0.42–1.21)	0.2129	1.24 (0.95–1.62)	0.1191	0.81 (0.62–1.06)	0.1252	1.34 (1.08–1.65)	0.0069	0.91 (0.73–1.12)	0.3677
Palate	2.17 (1.26–3.73)	0.0053	1.95 (1.13–3.36)	0.0167	1.99 (1.45–2.75)	<0.0001	1.42 (1.03–1.96)	0.0322	1.90 (1.46–2.47)	<0.0001	1.42 (1.09–1.85)	0.0090
Buccal	1.86 (1.32–2.64)	0.0005	1.73 (1.22–2.45)	0.0021	1.22 (1.00–1.50)	0.0483	0.98 (0.80–1.20)	0.8757	1.06 (0.90–1.24)	0.5191	0.91 (0.77–1.07)	0.2509
Retromolar	2.29 (1.53–3.43)	<0.0001	2.10 (1.40–3.15)	0.0003	1.49 (1.16–1.91)	0.0016	1.13 (0.88–1.45)	0.3272	1.24 (1.01–1.52)	0.0400	1.01 (0.82–1.24)	0.9566
Other sites	2.05 (1.30–3.23)	0.0019	1.87 (1.19–2.95)	0.0069	1.51 (1.14–2.00)	0.0039	1.08 (0.81–1.43)	0.6074	1.40 (1.12–1.77)	0.0038	1.08 (0.86–1.36)	0.5189
Tumor differentiation												
Well differentiated	1		1		1		1		1		1	
Moderately differentiated	1.22 (1.07–1.40)	0.0036	1.20 (1.05–1.38)	0.0089	1.62 (1.48–1.77)	<0.0001	1.24 (1.13–1.36)	<0.0001	1.51 (1.40–1.63)	<0.0001	1.22 (1.13–1.32)	<0.0001
Poorly differentiated	1.28 (1.03–1.59)	0.0231	1.19 (0.95–1.48)	0.1277	3.03 (2.69–3.41)	<0.0001	1.66 (1.47–1.88)	<0.0001	2.86 (2.59–3.17)	<0.0001	1.73 (1.56–1.92)	<0.0001

Other sites: tumors arising from the retromolar trigone or overlapping regions. Abbreviations: HR, hazard ratio; CI, confidence interval; S, surgery; CT, chemotherapy; RT, radiotherapy.

**Table 3 cancers-14-00015-t003:** Summary of published studies on the prognostic impact of surgical resection margins and associated clinical outcomes in oral malignancies.

Authors (Recruitment Year)	Patients (n)	Cutoff for Margins (mm)	Five-Year Outcome				Risk Factor (MVA)		
			LC (%)	DFS (%)	DSS (%)	OS (%)	LC	DSS	OS
Kang et al. (2011–2017) Current study	13768	>4/0.1–4/involved	92%/89%/87% (*p* < 0.0001)	nm	81%/76%/57% (*p* < 0.0001)	74%/67%/46% (*p* < 0.0001)	Margin, depth, site (palate/buccal/retromolar/other), differentiation	Margin, ENE, depth, p-stage, sex, age, pT, pN, site (palate), differentiation	Margin, ENE, depth, second primary, sex, age, pT, pN, site (palate), differentiation
Lin et al. [21] (2011–2017)	15654	≥5/4.0–4.9/3.0–3.9/2.0–2.9/1.0–1.9/0.1–0.9/involved	nm	nm	81%/81%/80%/77%/72%/75%/62% (*p* = nm)	76%/75%/74%/72%/66%/70%/47% (*p* = nm)	nm	Margin, sex, age, pT, pN, site (gum/palate), differentiation, treatment	Margin, sex, age, pT, pN, site (lip/palate), differentiation, treatment
Liao et al. [10] (1998–2005)	827	>7/≤7	89%/79% (*p* = 0.0003)	nm	82%/77% (*p* = 0.0287)	69%/63% (*p* = 0.0412)	Margin, depth, pN, betel quit	Margin, depth, alcohol, p-stage, ENE, differentiation	Margin, depth p-stage, ENE, differentiation
Binahmed et al. [12] (1975–2006)	425	≥2/<2/involved	27%/24%14%/ (*p* = 0.005)	nm	nm	68%/58%/39% (*p* < 0.0001)	nm	nm	nm
Jang et al. [13] (1996–2012)	325	>5/0–5	97%/85% (*p* = 0.002)	nm	nm	nm	Margin, pT, pN	nm	nm
		>5/involved	97%/25% (*p* < 0.001)	nm	nm	nm			
Nason et al. [14] (1975–2009)	554	≥5/3–4/≤2/involved	nm	71%/70%/49%/48% (*p* = 0.005)	nm	73%/70%/63%/39% (*p* < 0.0001)	nm	nm	nm
Zanoni et al. [16] (2000–2012)	381	>5/2.3–5.0/0.01–2.2/involved	nm	nm	nm	nm	Margin, PNI, pN	nm	nm
Mitchell et al. [21] (97 databases)	591	≥5/ 2–5/≤2 (involved)	nm	nm	nm	81%/75%/54% (*p* < 0.0001) *	nm	nm	nm
Loree et al. [2] (1979–1983)	398	≥5/<5	nm	nm	nm	60%/52% (*p* < 0.025)	nm	nm	nm

Abbreviations: MVA, multivariate analysis; LC, local control; DFS, disease-free survival; DSS, disease-specific survival; OS, overall survival; ENE, extra-nodal extension; PNI, perineural invasion; nm, not mentioned. * Chi-square test.

## Data Availability

Data sharing not applicable.

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
