# Peer review of "Surgical Margins Status and Prognosis after Resection of Oral Cavity Squamous Cell Carcinoma: Results from a Taiwanese Nationwide Registry-Based Study"

_cancers, 2021, doi:10.3390/cancers14010015_

Round 1
Reviewer 1 Report
An interesting multicenter study measuring prognosis outcome in patients affected by OSCC considering free margins after surgical removal. Although various similar studies are present in medical literature, given the considerable number of patients and this novel cut off at 4mm, the paper may be published after minor revisions; only some queries:
You said you excluded only 165 patients out of more than 34000...witch means that in situ carcinoma had a prevalence of 0,5%, which seems to me very low compared to my experience and medical literature....how do you explain that?
Also, you excluded more than 4000 patients for initial non surgical treatment, instead of patients that were not operable, which means that other therapies have been performed prior to surgery even if surgery was still possible? Please Specify
Page 1 line 85 you should add: "Various other topical and systemical therapies have been proposed in the management of this condition, with poor results on relapse rate and disease-free survival" and cite some articles such as: doi: 10.3390/curroncol28040213. and doi: 10.3390/medicina57060563.
Reviewer 2 Report
I read with interest the manuscript entitled "Surgical margins status and prognosis after resection of oral cavity squamous cell carcinoma: Results from a Taiwanese nationwide registry-based study".
Point of strenght is of course the capability to analyze data of a study population of more than 13.000 patients with known margin status after primary surgical resection of oral cavity SCC.
The manuscript is well written.
Point of weakness is the fact that there were statistically significant differences in terms of many important prognostic factors between the three groups of cutoffs (0-0.1/4-4 mm). In particular, there was a statistically significant difference in terms of treatment modality, which is important to be considered when evaluating the results.
I would like to have a comparison among margins equal or higher than 4 and margins equal or higher than 5 mm (as NCCN recommends). This would be important to underline that 4 mm is enough.
Moreover, if we consider NCCN guidelines, 4 mm would be considered as close surgical margins (= adverse feature) and adjuvant radiotherapy should be discussed by the MDT group. Could you please specify what are your indication for adjuvant treatment in the methods subheading? I think this would add more clarity to your findings.
Reviewer 3 Report
The question of adequate margins in cancer surgery is continuously under discussion. Current guideline is >5mm but several other cut off measures have suggested as discussed also in the discussion of this manuscript. To my best knowledge, the problem with all the published margin studies is that patients are not treated with surgery only but a variable proportion of the patients in different groups with variable margins have received additional therapy (RT or CRT). This is the main weakness of this study as well. The second point of criticism is the selected cut off value >4mm. It is understandable that 4mm was taken according to cox proportional hazard model, but in the real life >4mm and >5mm do not differ that much.
In this study there was 6749 patients who were treated with surgery alone. This population should be analysed regarding surgical margins and the outcome. In addition to this information the whole patient population of 13768 can also be reported but I would suggest a change in the cut off value. If the data would be analysed with cut off >4mm it would make sense and difference comparing to current guideline. As I already mentioned in the real life >4mm and >5mm are almost the same. Secondly, involved margin is usually up to 1mm (not 0.1mm) and this should be changed in this study.
Round 2
Reviewer 3 Report
For some reason in the editing of my first response there has been a problem as the marking “equal or more ≥” is the same as “more >”. Therefore I am not surprised my suggestions have not been understood correctly. My point was that the authors should include 4mm into their cutoff value of clear margin and make the analysis based on three categories: Involved (0-0.99mm) – close (1-3.99mm) – clear ≥4mm (equal or more than 4). If one does not include 4mm into the cutoff then it is not much new comparing to present ≥5mm. You can ask any pathologist.
Secondly the authors say they have done the analyses of patients treated with surgery alone but they have not added this into the manuscript. This is needed as it is the only way of knowing what margin is enough for not adding adjuvant treatment.
Author Response
Thank you for your comments and clarifications. As requested, we reanalyzed the margin status according to the following categories: 0−0.99 mm, 1−3.99 mm, and ≥4 mm. The following 5-year outcomes were observed in the three margin groups: local control, 88%, 88%, and 92% (P <0.001); disease-specific survival, 62%, 74%, and 81% (P <0.001); and overall survival, 52%, 65%, and 74% (P <0.001), respectively. These figures are consistent with those reported in the paper using the following categorization: 0 mm, 0.1−4 mm, and >4 mm. As per suggestion, we conducted additional calculations and provided the results according to the proposed classification (0−0.99 mm, 1−3.99 mm, and ≥4 mm) in the attached Supplementary File. Using the 5-year local control as the outcome of interest, we constructed a multivariable Cox proportional hazard regression model with a spline function to test the predictive value of different cutoff points (from 0 to 30 mm). A margin of 4 mm produced a hazard ratio of 1.0245. In this scenario, we decided to adopt a cutoff of less than or equal to 4 mm. As for the data pertaining to patients treated with surgery alone, they have been reported in the revised paper (from line 139 to line 150 and from line 281 to line 285, text highlighted in orange color).

Round 3
Reviewer 3 Report
Thank you for adding the requested data.